# Eps15 Homology Domain Protein 4 (EHD4) is required for Eps15 Homology Domain Protein 1 (EHD1)-mediated endosomal recruitment and fission

Tyler Jones[1], Naava Naslavsky[1], Steve Caplan[1,2]*

**1** Department of Biochemistry & Molecular Biology, University of Nebraska Medical Center, Omaha, NE, United States of America, **2** Fred and Pamela Buffett Cancer Center, University of Nebraska Medical Center, Omaha, NE, United States of America

* scaplan@unmc.edu

**Data Availability Statement:** All relevant data are within the manuscript and its Supporting Information files.

## Abstract

Upon internalization, receptors are trafficked to sorting endosomes (SE) where they undergo sorting and are then packaged into budding vesicles that undergo fission and transport within the cell. Eps15 Homology Domain Protein 1 (EHD1), the best-characterized member of the Eps15 Homology Domain Protein (EHD) family, has been implicated in catalyzing the fission process that releases endosome-derived vesicles for recycling to the plasma membrane. Indeed, recent studies suggest that upon receptor-mediated internalization, EHD1 is recruited from the cytoplasm to endosomal membranes where it catalyzes vesicular fission. However, the mechanism by which this recruitment occurs remains unknown. Herein, we demonstrate that the EHD1 paralog, EHD4, is required for the recruitment of EHD1 to SE. We show that EHD4 preferentially dimerizes with EHD1, and knock-down of EHD4 expression by siRNA, shRNA or by CRISPR/Cas9 gene-editing leads to impaired EHD1 SE-recruitment and enlarged SE. Moreover, we demonstrate that at least 3 different asparagine-proline-phenylalanine (NPF) motif-containing EHD binding partners, Rabenosyn-5, Syndapin2 and MICAL-L1, are required for the recruitment of EHD1 to SE. Indeed, knock-down of any of these SE-localized EHD interaction partners leads to enlarged SE, presumably due to impaired endosomal fission. Overall, we identify a novel mechanistic role for EHD4 in recruitment of EHD1 to SE, thus positioning EHD4 as an essential component of the EHD1-fission machinery at SE.

## Introduction

Upon internalization, receptors, lipids and extracellular fluid are segregated into budding vesicles that are cleaved from the plasma membrane and trafficked to a key endocytic compartment known as the early or sorting endosome (SE) [1]. The SE is a key sorting organelle, and from this organelle, receptors may be transported to late endosomes and lysosomes for degradation, or alternatively, recycled back to the plasma membrane for additional rounds of

**Funding:** S.C. was supported by R01 GM123557 from the National Institutes of General Medical Sciences at the National Institutes of Health. The authors thank the University of Nebraska Medical Center Advanced Microscopy Core Facility, which receives partial support from NIGMS: INBRE P20 GM103427 and COBRE P30 GM106397 grants, as well as support from the National Cancer Institute for The Fred & Pamela Buffett Cancer Center Support Grant P30 CA036727, and the Nebraska Research Initiative. The funders had no role in study design, data collection, analysis, decision to publish or manuscript preparation.

**Competing interests:** The authors have declared that no competing interests exist.

internalization [2]. In recent years, significant advances have been made in understanding the complex mechanisms that regulate cargo sorting at the SE. For example, it was demonstrated that the ARF GTPase activating protein (GAP), ARF GAP with coiled-coil ankyrin repeat and PH domain-containing protein 1 (ACAP1), interacts with a variety of receptors to direct them back to the plasma membrane [3]. Moreover, coupled with the retromer complex, which entails a Cargo Selection Complex (CSC) trimer of VPS35, VPS29 and either a VPS26a or VPS26b isoform, along with a dimer of sorting nexins (SNX1 or SNX2, and SNX5, SNX6 or SNX32) [4, 5], two members of the sorting nexin family, SNX17 and SNX27, have recently been implicated in controlling the recycling of multiple receptors via interactions between their FERM domains and the cytoplasmic tails of the receptors [3, 6–9]. In addition, the involvement of the SNX17-associated retriever and CCC complexes [10, 11] have further highlighted the active and complex mechanisms by which proteins are sorted and recycled to the plasma membrane.

Despite the progress in understanding the players and mechanisms of sorting at the SE, how the fission of budding vesicles at the SE occurs remains poorly understood. Upon incorporation of receptors into budding transport vesicles, it is necessary to recruit fission machinery for vesicle release. In addition to the retromer [12], the Wiskott–Aldrich syndrome protein (WASH) complex, comprised of WASH1 (also known as WASHC1), Strumpellin (WASHC5), CCDC53 (WASHC3), KIAA1033/SWIP (WASHC4) and Fam21 (also known as WASHC2) [13, 14] has been implicated in vesicular fission via actin nucleation [15]. A recent study further suggest involvement of a novel complex including Rab11-FIP5, VIPAS39, VPS45, Rabenosyn-5 and the dynamin-like Eps15 Homology Domain protein, EHD1 [16]. However, the potential involvement of dynamin-like proteins such as EHD1 and nucleotide hydrolysis at the SE remains likely.

Over the past two decades, the Eps15 Homology Domain protein 1 (EHD1) has emerged as a major regulator of endocytic recycling [17, 18]. Recent studies have demonstrated that in vitro, EHD1 is capable of membrane fission, whereas in cells it localizes to SE and recycling endosomes, and induces ATP-catalyzed membrane fission [19–26]. We have recently demonstrated that EHD1 undergoes recruitment to endosomal membranes upon induction of receptor-mediated endocytosis, where it interacts with SNX17 and promotes endosomal fission [27]. In addition, a number of key asparagine-proline-phenylalanine motif-containing endosomal proteins have been identified that interact with EHD1 and/or EHD4, and may serve to recruit the latter to endosomes [18, 28], including Rabenosyn-5 [29], Syndapins [30, 31], MICAL-L1 [25], and others. While our data are consistent with a model in which SNX17 and EHD1 couple endosomal sorting and the endosomal fission machinery, the mechanism of EHD1 recruitment to endosomes remains unclear.

Herein, we address the potential role of the EHD1 paralog, EHD4, in the process of EHD1 endosomal recruitment and fission. EHD4 shares ~70% identity with EHD1, and has been characterized as a potential EHD1 interaction partner and regulator of trafficking from SE [32, 33]. We demonstrate that EHD4 hetero-dimerizes with EHD1, apparently displaying higher propensity for hetero-dimerization than homo-dimerization, suggesting that it may contribute to the regulation of EHD1 recruitment to endosomes. Consistent with this notion, we showed that impaired EHD4 expression, via siRNA, shRNA or CRISPR/Cas9 gene-editing all led to decreased EHD1 recruitment to endosomes. Moreover, EHD4-depleted cells displayed enlarged SE, likely resulting from impaired endosomal fission. Finally, we demonstrated that EHD4 shares several key endosomal binding partners with EHD1, including Rabenosyn-5 and Syndapin2, and their depletion similarly leads to reduced EHD1 endosomal recruitment and fission. Our findings recognize EHD4 as an important regulator of EHD1-mediated endosomal recruitment and fission.

## Materials and methods

### Cell lines

The HeLa cervical cancer cell line was acquired from the American Type Culture Collection (ATCC, CRM-CCL-2). NIH3T3 (ATCC, CRL-1658) parental cells were subjected to CRISPR/Cas9 to generate the NIH3T3 cell line expressing endogenous levels of EHD1 with GFP attached to the C-terminus, as well as the EHD1 knock-out, EHD4 knock-out and EHD1/EHD4 double knock-out cells as described [34, 35]. Both HeLa and NIH3T3 cells were cultured at 37˚C in 5% CO2 in DMEM/High Glucose (HyClone, SH30243.01) containing 10% heat-inactivated Fetal Bovine Serum (Atlanta Biologicals, S1150), 1x Penicillin Streptomycin (Gibco, 15140–122), 50 mg of Normocin (InvivoGen, NOL-40-09), and 2 mM L-Glutamine (Gibco, 25030–081). All cell lines were routinely tested for Mycoplasma infection.

### Antibodies

The following antibodies were used: Rabbit anti-EHD1 (Abcam, ab109311), Rabbit anti-EHD4 [32], Rabbit anti-EEA1 (Cell Signaling, #3288), Rabbit anti-HA (SAB, #T501), Mouse anti-GFP (Roche, 11814460001), Mouse anti-LRP1 (Novus, NB100-64808), Donkey anti-mouse-HRP (Jackson, 715-035-151), Mouse anti-rabbit IgG light chain-HRP (Jackson, 211-032-171), Alexa-fluor 568-conjugated goat anti-rabbit (Molecular Probes, A11036).

### DNA constructs, cloning, and site-directed mutagenesis

Cloning of PTD1, EHD1, EHD4, MICAL-L1, Rabenosyn-5 1–263, and Rabenosyn-5 151–784 in the yeast two-hybrid vector pGADT7 and cloning of PVA3, EHD1, EHD4, EHD1 V203P, EHD1 aa1-439, EHD1 aa1-309, EHD1 aa1-199, EH-1, EHD2, EHD3, EHD3Δcc, MICAL-L1, and Syndapin2 in the yeast two-hybrid vectors were described previously [31, 32, 36–39]. The following constructs were generated via site-directed mutagenesis using Q5 High-Fidelity 2X Master Mix (New England Biosciences, M0492S) according to the manufacturer's protocol: pGADT7-EHD4 S522A, pGADT7-EHD4 S523D, and pGADT7-EHD4 SS522AD.

### Co-immunoprecipitation

HeLa cells were cultured on 100 mm plates until confluent. Cells were lysed with lysis buffer made from 50 mM Tris, pH 7.4, 100 mM NaCl, 0.5% Triton X-100, and 1 x protease cocktail inhibitor (Millipore, 539131) on ice for 15 min with mixing every 5 min. Lysates were centrifuged to clear insoluble matter and then incubated in the absence of antibody or with rabbit anti-EHD1 (Abcam, ab109311) overnight on a rotator at 4˚C. Protein G Sepharose Beads 4 Fast Flow (GE Healthcare, 17-0618-01) were added to both control and antibody-containing lysates and mixed on a rotator at 4˚C for 4 h. Samples were then washed with the aforementioned lysis buffer and centrifuged at 22,000 x g at 4˚C for 30 s. Washes were performed a total of three times and proteins were eluted from the beads by boiling in 4x loading buffer (250 mM Tris, pH 6.8, 8% SDS, 40% glycerol, 5% β-mercaptoethanol, 0.2% bromophenol blue) for 10 min and detected by immunoblotting.

### Yeast two-hybrid assay

AH109 yeast were cultured overnight in YPD media containing 10 g/L Bacto Yeast Extract (BD, Ref. 212750), 20 g/L Peptone (Fisher Scientific, CAS RN: 73049-73-7, BP1420-500), and 20 g/L Dextrose (Fisher Scientific, CAS RN: 50-99-7, BP350-1) at 30˚C and 250 RPM. Cultures were then spun down at 975 x g for 5 min and the supernatant was aspirated. Pellets were rinsed with autoclaved MilliQ water and centrifuged for an additional 5 min at 975 x g and the supernatant was aspirated. Pellets were resuspended in a suspension buffer of 80% autoclaved

MilliQ water, 10% lithium acetate pH = 7.6, and 10% 10x TE pH = 7.5. 125 µl aliquots of the cell suspension were then incubated each with 600 µl of PEG solution (40% PEG (CAS RN: 25322-68-3, Prod. Num. P0885), 100 mM lithium acetate pH = 7.6 in TE pH = 7.5). 1 µl of Yeastmaker Carrier DNA (TaKaRa Cat# 630440) was added to each aliquot, followed by 1 µg of each respective plasmid, and mixed by inverting twice, then by vortexing twice. Mixtures were then incubated at 30˚C and 250 RPM for 30 min. 70 µl of DMSO was added to each tube, followed by inverting/mixing twice, and mixtures were placed at 42˚C for 1 h. Samples were then placed on ice for 5 min, followed by centrifugation at 22,000 x g for 30 s. The supernatant was aspirated and the samples were resuspended in 40 µl of autoclaved MilliQ water. 15 µl of each sample was then plated and spread on -2 plates (+His) made using 27 g/L DOB Medium (MP, Cat. No. 4025–032), 20 g/L Bacto Agar (BD, Ref. 214010), and 0.64 g/L CSM-Leu-Trp (MP, Cat. No. 4520012) and incubated at 30˚C for 72–96 h. Following the incubation period, three separate colonies from each sample were selected and added to 600 µl of autoclaved MilliQ water. In a clean cuvette, 500 µl of the mixture was added to 500 µl of autoclaved water and measured using a spectrophotometer at 600 nm. Mixtures were then normalized to 0.100 λ and 15 µl of each mixture was spotted onto both a -2 plate and a -3 plate (-His) made using 27 g/L DOB Medium (MP, Cat. No. 4025–032), 20 g/L Bacto Agar (BD, Ref. 214010), and 0.62 g/L CSM-His-Leu-Trp (MP, Cat. No. 4530112). Both plates were incubated at 30˚C for 72 h and imaged.

## siRNA treatment

CRISPR/Cas9 gene-edited NIH3T3 cells expressing endogenous levels of EHD1 with GFP fused to the C-terminus were plated on fibronectin-coated coverslips and grown for 4 h at 37˚C in 5% CO2 in DMEM/High Glucose (HyClone, SH30243.01) containing 10% heat inactivated Fetal Bovine Serum (Atlanta Biologicals, S1150), 1x Penicillin Streptomycin (Gibco, 15140–122), 50 mg of Normocin (InvivoGen, NOL-40-09), and 2 mM L-Glutamine (Gibco, 25030–081). The cells were then treated with either mouse EHD4 siRNA oligonucleotides (Dharmacon, Custom Oligonucleotide, Seq: GAGCAUCAGCAUCAUCGACdTdT), mouse Rabenosyn-5 siRNA oligonucleotides (Dharmacon, On-TARGETplus SMARTpool, cat # L-056534-01-0010), mouse Syndapin 2 siRNA oligonucleotides (Dharmacon, On-TARGETplus SMARTpool, cat # L-045093-01-0005), or mouse MICAL-L1 siRNA oligonucleotides (Dharmacon, On-TARGETplus SMARTpool, cat # L-049952-00-0005) for 72 h at 37˚C in 5% CO2 in 1x Opti-MEM 1 (Gibco, 31985–070) containing 12% heat inactivated Fetal Bovine Serum (Atlanta Biologicals, S1150) and 2 mM L-Glutamine (Gibco, 25030–081) using Lipofectamine RNAiMax transfection reagent (Invitrogen, 56531), following the manufacturer's protocol.

## shRNA treatment

HeLa cells were plated on coverslips and grown for 24 h at 37˚C in 5% CO2 in DMEM/High Glucose (HyClone, SH30243.01) containing 10% heat inactivated Fetal Bovine Serum (Atlanta Biologicals, S1150), 1x Penicillin Streptomycin (Gibco, 15140–122), 50 mg of Normocin (InvivoGen, NOL-40-09), and 2 mM L-Glutamine (Gibco, 25030–081). The cells were then treated with pLKO.1-EHD4 shRNA [40] for 72 h at 37˚C in 5% CO2 in DMEM/High Glucose (HyClone, SH30243.01) containing 10% heat inactivated Fetal Bovine Serum (Atlanta Biologicals, S1150), and 2 mM L-Glutamine (Gibco, 25030–081), using FuGene 6 Transfection Reagent (Promega, E2691), following the manufacturer's protocol.

## Transfection

CRISPR/Cas9 gene-edited NIH3T3 cells expressing endogenous levels of EHD1 with GFP fused to the C-terminus were plated on fibronectin-coated coverslips and grown for 4 h at

37˚C in 5% CO2 in DMEM/High Glucose (HyClone, SH30243.01) containing 10% heat inactivated Fetal Bovine Serum (Atlanta Biologicals, S1150), 1x Penicillin Streptomycin (Gibco, 15140–122), 50 mg of Normocin (InvivoGen, NOL-40-09), and 2 mM L-Glutamine (Gibco, 25030–081). The cells were then treated with HA-tagged EHD4 in pcDNA 3.1 (+) (Invitrogen, V79020) for 72 h at 37˚C in 5% CO2 in DMEM/High Glucose containing 10% heat inactivated Fetal Bovine Serum, and 2 mM L-Glutamine, using FuGene 6 Transfection Reagent (Promega, E2691), following the manufacturer's protocol.

## Immunofluorescence and LRP1 uptake

CRISPR/Cas9 gene-edited NIH3T3 cells expressing endogenous levels of EHD1 with GFP attached to the C-terminus were subjected briefly to LRP1 uptake. Uptake was performed by diluting Mouse anti-LRP1 (Novus, NB100-64808) (1:70) in DMEM/High Glucose (HyClone, SH30243.01) containing 10% heat inactivated Fetal Bovine Serum (Atlanta Biologicals, S1150), 1x Penicillin Streptomycin (Gibco, 15140–122), 50 mg of Normocin (InvivoGen, NOL-40-09), and 2 mM L-Glutamine (Gibco, 25030–081) in an ice water bath for 30 min, followed by 2 washes with 1x PBS. Pre-warmed DMEM media, as previously described, was added to these coverslips, which were incubated at 37˚C in 5% CO2 for 30 min and washed once in 1x PBS. Following treatment, cells were then fixed in 4% paraformaldehyde (Fisher Scientific, BP531-500) in PBS for 10 min at room temperature. After fixation, cells were rinsed 3 times in 1x PBS and incubated with primary antibody in staining buffer (1x PBS with 0.5% bovine serum albumin and 0.2% saponin) for 1 h at room temperature. Cells were washed 3 times in 1x PBS, followed by incubation with the appropriate fluorochrome-conjugated secondary antibody diluted in staining buffer for 30 min. Cells were washed 3 times in 1x PBS and mounted in Fluoromount-G (SouthernBiotech, 0100–01). Z-stack confocal imaging was performed using a Zeiss LSM 800 confocal microscope with a 63x/1.4 NA oil objective. 10 fields of cells from each condition were collected from 3 independent experiments and assessed using NIH ImageJ.

## Graphical and statistical analysis

NIH ImageJ was used to quantify particle count, total area, average size, % area, mean, and integral density. Size parameters were set for 0 –infinity. Circularity parameters were set to 0.0–1.0. Brightness parameters were set to either 75–255 or 125–255, though calculations for 100–255, 150–255, and 175–255 were also conducted. Brightness parameters were selected to eliminate recognition of background by ImageJ's particle counter while optimizing selection of true positive fluorescent pixels. All statistical analyses were performed with significance using an independent sample two-tailed t-test under the assumption that the two samples have equal variances and normal distribution using the Vassarstats website (http://www.vassarstats.net/), or when comparing multiple samples, with a one-way ANOVA using post-hoc Tukey test for significance (https://astatsa.com). To address biological variations between individual tests, we have designed a modified version of the method described by Folks [41] for deriving a "consensus p-value" to determine the likelihood that the collection of different test/experiments *collectively suggests* (or refutes) a common null hypothesis, modified from the Liptak-Stouffer method [42]. All the graphics were designed using GraphPad Prism 7.

## Results and discussion

Given the role of EHD1 in the regulation of endocytic recycling [21, 24, 25, 29, 36], and its sequence identity and relationship with EHD4 [32, 43], as well as the recent evidence supporting a role for EHD1 in endosomal fission [19, 20, 22, 23, 27, 44–46], we hypothesized that EHD4 coordinates fission and recycling with EHD1. To first test this idea, we assessed the

ability of endogenous EHD1 and EHD4 to co-immunoprecipitate. As demonstrated in Fig 1A (left panel), EHD4 appeared in the cell lysates as a ~64 kDa band, and a band of the same size was co-immunoprecipitated by antibodies against EHD1, but not a beads-only control. Although we previously tested our EHD4 antibodies and demonstrated that they do not recognize EHD1 [32], to ensure that the ~64 kDa band was indeed EHD4 and not cross-recognition of EHD1 by the EHD4 antibody, we stripped the nitrocellulose filter paper and reblotted with antibodies to EHD1 (Fig 1A; right panel). Blotting with anti-EHD1 led to detection of a faster migrating ~60 kDa band that clearly migrated below the ~64 kDa EHD4 band, demonstrating that EHD1 and EHD4 reside in a complex in cells. Moreover, consistent with previous findings [32, 33], we found that HA-tagged EHD4 displayed partial co-localization (Pearson's Coefficient 0.677 with standard deviation of 0.048) with EHD1-GFP in our CRISPR/Cas9 NIH 3T3 gene-edited cells expressing EHD1-GFP (S1 Fig).

We further analyzed the nature of EHD1-EHD4 interactions by instituting a series of truncations and/or mutations in EHD1 (Fig 1B) and testing whether it hetero-dimerizes with EHD4. As shown by yeast two-hybrid (Y2H) analysis, EHD1 both homo-dimerizes and hetero-dimerizes with EHD4, whereas EHD4 preferentially hetero-dimerizes with EHD1, but displays little propensity to homo-dimerize (Fig 1C). In addition, whereas the EH domain of EHD proteins is a well-characterized protein-binding module [28, 47–49], it remains superfluous for EHD dimerization. Since the structure of EHD1 is organized via several domains in addition to the C-terminal EH domain (Fig 1A), we addressed the role of these domains through a series of truncations and mutations. Indeed, truncations in the second helical domain (residue 309) or within the ATP-binding domain (residue 199) abrogated dimerization, whereas a coil-breaking valine-to-proline (V203P) in EHD1 [29] led to dramatically reduced EHD1-EHD4 hetero-dimerization, but had little effect on EHD1 homo-dimerization (Fig 1C). Moreover, EHD4 hetero-dimerized with EHD3 (which displays 86% identity with EHD1) and this interaction was similarly abrogated by the EHD3 V203P coil-breaking mutant (Fig 1D). However, EHD4 was unable to hetero-dimerize with EHD2 (which displays only 67% identity with EHD1 and diverges significantly from the functions of its other paralogs [50–60]) (Fig 1D).

Given the role of EHD1 in endosomal fission [19–26] and our previous study suggesting that EHD4 regulates endosomal size [32], we hypothesized that EHD4 might coordinate endosome fission and size with EHD1. To test this idea, we took untreated cells, mock-shRNA transfected cells, or cells subjected to EHD4-shRNA knock-down and examined EEA1-labeled SE after immunostaining (Fig 2A–2F). As demonstrated, EHD4-shRNA reduced EHD4 levels to almost non-detectable, with only a slight effect on EHD1 levels, potentially due to destabilization coming from the loss of hetero-dimerization (Fig 2G). While the EEA1-labeled SE were generally homogeneous in size and distribution in the untreated and mock-transfected cells (Fig 2; compare 2A and the inset in 2D to 2B and the inset in 2E), SE displayed a significant increase in size upon acute EHD4-depletion (Fig 2C; inset in 2F). Indeed, quantification of mean EEA1-labeled SE size demonstrated a 2-3-fold increase in the acute absence of EHD4 (Fig 2H), suggesting that EHD4 regulates endosomal fission, potentially through its interaction with EHD1.

As we have recently demonstrated that EHD1-depletion impairs fission and induces enlarged SE [27], we next asked whether simultaneous depletion of EHD1 and EHD4 further impedes endosomal fission and leads to increased SE size. To this aim, we used CRISPR/Cas9 NIH3T3 cells that were gene-edited and chronically lack EHD1 (EHD1 KO), EHD4 (EHD4 KO) or both EHD1 and EHD4 (EHD1/EHD4 DKO) [34, 35] (Fig 3; protein expression validated in 3I). As demonstrated, in both the EHD1 and EHD4 single KO cell lines, SE size was modestly but significantly larger than in the wild-type parental NIH3T3 cell line (compare B

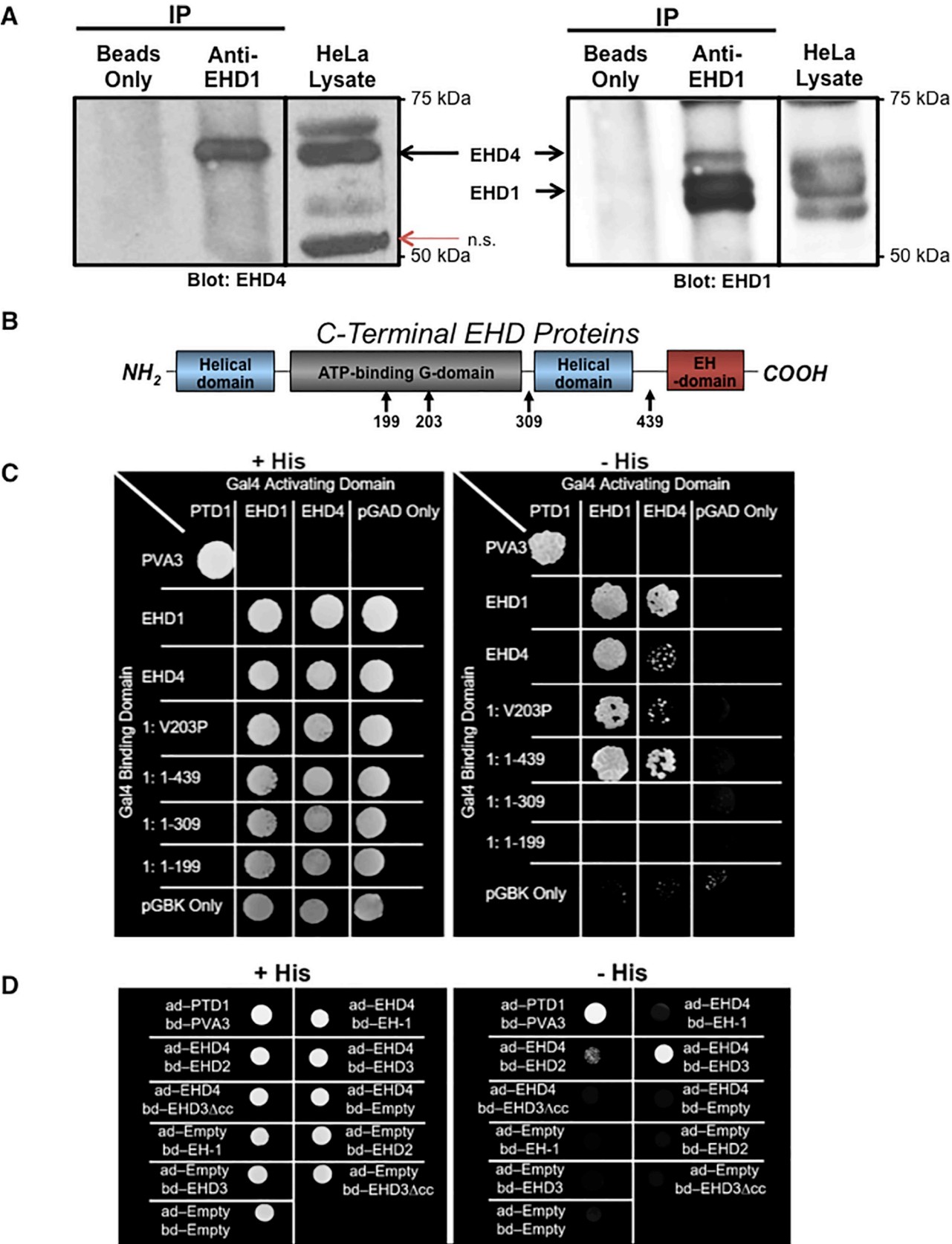

**Fig 1. Interaction between endogenous EHD1 and EHD4.** *A*, EHD4 co-immunoprecipitates with EHD1. HeLa lysates were incubated at 4˚C overnight in the presence or absence of anti-EHD1 antibody. Protein G beads were then added to the lysate-only (beads only) or lysate-antibody (anti-EHD1) mix at 4˚C for 3 h. Bound proteins were then eluted by boiling for 10 min at 95˚C in β-mercaptoethanol-containing loading buffer, separated by SDS-PAGE, and immunoblotted with anti-EHD4 antibodies (left panel) or stripped and then immunoblotted with anti-EHD1 antibodies (right panel). Input lysates (25%) are depicted (left and right panels, right lane). *B*, Schematic diagram depicting the domain architecture of C-terminal EHD proteins, indicating residues that were replaced by site-directed mutagenesis and identifying the truncations used in this study. *C*, Yeast two-hybrid colony growth characterizing the interaction between EHD1 and EHD4. Co-transformation of both pGADT7 and pGBKT7 vectors is required for growth on plates lacking leucine and tryptophan (left panel; +His plates), whereas interaction between the fusion proteins is required for histidine synthesis and growth on -His plates (right panel). + His plates illustrate that both target vectors have been transformed into the yeast.—His plates inform whether the proteins of interest interact. 1: V203P represents an amino acid substitution at residue 203 of EHD1 that is predicted to interfere with coiled-coil formation, whereas 1: 1–439, 1: 1–309 and 1: 1–199 represent various EHD1 truncations. *D*, Yeast two-hybrid colony growth characterizing the interaction between EHD4 and other EHD proteins. EHD3Δcc represents full-length EHD3 with a valine to proline substitution at residue 203, whereas EH-1 represents the EHD1 EH domain only (residues 436–534). n.s.; non-specific band, ad; activation domain, bd; binding domain.

and the inset in F, and C and the inset in G, to A and the inset E, and quantified in J). Moreover, in the EHD1/EHD4 DKO cell line, an additional increase in EEA1-labeled SE size was noted (Fig 3D and inset in 3H, and quantified in 3J). It is of interest that acute siRNA knockdown of EHD4 induces much larger endosome size than the more chronic EHD4 knock-out in the CRISPR/Cas9 NIH3T3 cells, suggesting that compensation may be occurring in the latter cells. Overall, these data are consistent with the role for EHD1 in vesiculation of tubular and vesicular recycling endosomes [19, 20, 22, 23, 45] and further support the notion that both EHD1 and EHD4 regulate endosomal fission, as their depletion leads to enlarged SE in cells.

We have recently demonstrated that upon stimulation of receptor-mediated endocytosis, EHD1 can be recruited to SE to carry out fission and facilitate cleavage of budding vesicles and endocytic recycling [27]. However, despite the homology between the 4 EHD paralogs [18, 61], thus far only EHD1 has been directly implicated in fission. Accordingly, based on the interactions we characterized between EHD1 and EHD4, we hypothesized that EHD4 may be required for the recruitment of EHD1 to SE. To address this, we incubated CRISPR/Cas9 gene-edited NIH3T3 cells that express EHD1-GFP with antibodies to low-density lipoprotein receptor-related protein 1 (LRP1) to induce internalization of the receptor-antibody complexes (Fig 4). We have previously demonstrated that by inducing uptake of receptors such as LRP1 or transferrin receptor (unpublished observations) for 30 minutes, we can observe 2–3 fold increases in the recruitment of cytoplasmic EHD1 to SE [27]. The cells were either untreated (A and inset in D), mock-treated (B and inset in E) or EHD4-depleted with siRNA (C and inset in F). EHD4 siRNA knock-down was validated by immunoblotting, and EHD1-GFP levels were similar in untreated, mock and cells where EHD4 knock-down was effected by siRNA (Fig 4G). As demonstrated, in the untreated and mock-treated cells, LRP1 uptake led to the localization of EHD1-GFP to a smattering of vesicular and short tubular SE (Fig 4A and inset in 4D, 4B and inset in 4E; quantified in 4H). However, EHD4 knock-down led to a significant decrease in the number of vesicular and tubular SE marked by EHD1-GFP (Fig 4C and inset, 4F; quantified in 4H). These data suggest that EHD4 is required for the optimal recruitment of EHD1 to SE upon receptor-mediated internalization.

EHD1 has been characterized as a protein that binds to motifs containing the tripetide asparagine-proline-phenylalanine (NPF) through its Eps15 homology (EH) domain, particularly when the motif is followed directly by negatively charged residues [28, 47–49, 62, 63]. However, its closest paralog, EHD3, displays more restricted binding to interaction partners [44], and the binding selectivity of EHD4 has not been well characterized. Accordingly, we hypothesized that EHD4 may interact with a subset of NPF-containing proteins that localize to SE and help anchor EHD1-EHD4 dimers to the cytoplasmic side of the SE membrane upon receptor-mediated internalization. To test this notion, we first assessed whether EHD4 could interact with several of the key EHD1-binding partners that contain NPF motifs and localize

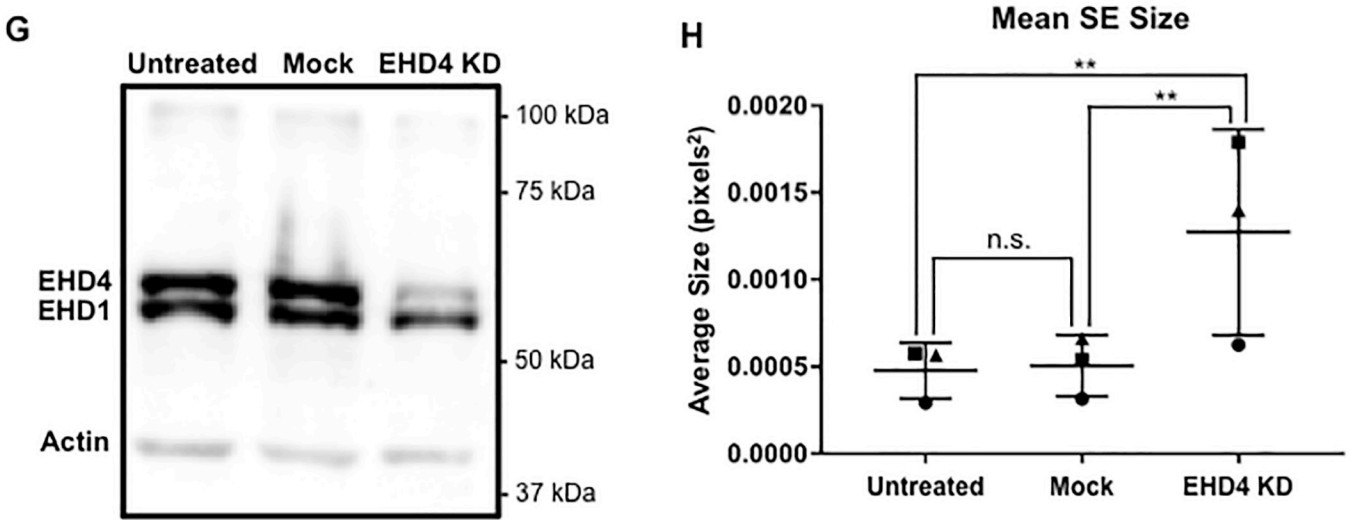

**Fig 2. EHD4 depletion induces enlarged sorting endosomes.** *A-F*, Representative micrographs and insets depicting EEA1-labeled endosomes in untreated, mock-treated, and EHD4 knock-down cells. HeLa cells were either untreated (*A* and inset in *D*), mock-treated with transfection reagent (*B* and inset in *E*), or transfected with an EHD4 shRNA construct (*C* and inset in *F*) for 72 h, fixed and immunostained with an EEA1 antibody prior to imaging. *G*, Validation of EHD4 shRNA efficacy by immunoblot analysis. *H*, Graph depicting differences in mean EEA1-labeled SE size in untreated, mock-treated and EHD4 knock-down cells. Error bars denote standard deviation and p-values for each experiment were determined by one-way ANOVA for individual experiments using a post-hoc Tukey HSD calculator to determine significance. All 3 experiments rely on data from 10 images and each experiment is marked by a distinct shape on the graph. Significance between samples for the 3 experiments was calculated by deriving a consensus p-value (see Materials and methods). Micrographs are representative orthogonal projections from three independent experiments, with 10 sets of z-stacks collected for each treatment per experiment. Bar, 10 μm. n.s. = not significant (p > 0.5), **p < 0.00001.

to SE. Initially, we used Y2H to assess interactions between EHD4 and both Rabenosyn-5 [29] and MICAL-L1, the latter which recruits EHD1 not only to endosomes [25, 31, 64] but also to the centrosome [65]. As demonstrated, similar to EHD1, EHD4 interacted with Rabenosyn-5 (Fig 5A). Somewhat surprisingly, despite being able to interact with EHD1, MICAL-L1 did not display binding to EHD4 (Fig 5A). We have previously shown that an alanine-aspartic acid pair within the EHD1 EH domain (at residues 519 and 520) was required for its selective binding to Rabankyrin-5, whereas the other EHD paralogs did not bind Rabankyrin-5 [44]. We also demonstrated that mutation of the asparagine-glutamic acid pair at the same residues (519 and 520) in the EHD3 EH domain to the alanine-aspartic acid pair found in that position in EHD1 altered its binding selectivity and facilitated an interaction with Rabankyrin-5 [44]. Accordingly, we now asked whether mutation of EHD4's serine-serine to alanine-aspartic acid at the position that aligns with EHD1's alanine-aspartic acid residues (residues 519 and 520) would allow it to bind to MICAL-L1 (Fig 5B and 5C). However, as demonstrated, EHD4 remained unable to interact with MICAL-L1, even after the SS to AD substitutions. Since we have shown that MICAL-L1 binds to another NPF-containing protein, Syndapin2 [31], we also tested EHD4-Syndapin2 binding (Fig 5D). As shown, both EHD1 and EHD4 were able to bind Syndapin2. These data suggest that EHD1 homo-dimers and EHD1-EHD4 hetero-dimers have multiple potential recruitment targets on SE.

If the EHD1 and EHD4 NPF-containing binding partners are required for EHD1 recruitment to SE and subsequent fission, we rationalized that their depletion would cause reduced EHD1-GFP recruitment to SE upon stimulation of receptor-mediated internalization and increased endosomal size due to impaired fission. Accordingly, we first knocked-down Rabenosyn-5, a potential SE binding partner for both EHD1 and EHD4 (Fig 6). After validating Rabenosyn-5 knock-down efficacy (Fig 6M), we compared EEA1-labeled SE size and EHD1-GFP recruitment to SE in mock-treated cells (Fig 6A–6F) and Rabenosyn-5 knock-down cells (Fig 6G–6L) upon stimulation of receptor-mediated internalization. As demonstrated, upon Rabenosyn-5 knock-down, significantly less EHD1-GFP was observed on SE (compare Fig 6H and 6K to 6B and 6E; quantified in 6O). Moreover, EEA1-labeled SE were significantly larger in Rabenosyn-5 knock-down cells (compare Fig 6G and 6J to 6A and 6D; quantified in 6N). One possibility was that EHD4 helps mediate an interaction between Rabenosyn-5 and EHD1; however, upon EHD4 knock-down we still observed interactions between EHD1 and Rabenosyn-5 (S2 Fig), suggesting that this is not the case. Overall, these data support the notion that Rabenosyn-5 plays a role in the recruitment of EHD1 homo-dimers and EHD1-EHD4 hetero-dimers to SE.

We next tested whether Syndapin2, another EHD1 and EHD4 endosomal binding partner, was similarly required for EHD1 recruitment to SE and endosomal fission (Fig 7). Syndapin2 knock-down efficacy was first verified by immunoblotting (Fig 7M). As demonstrated, Syndapin2 knock-down led to dramatically reduced recruitment of EHD1-GFP to endosomes (compare Fig 7H and 7K to 7B and 7E; quantified in 7O). Indeed, impaired recruitment of EHD1-GFP led to enlarged EEA1-labeled SE (compare Fig 7G and 7J to 7A and 7D; quantified in 7N). These data suggest that Syndapin2 is involved in recruitment of EHD1 to SE.

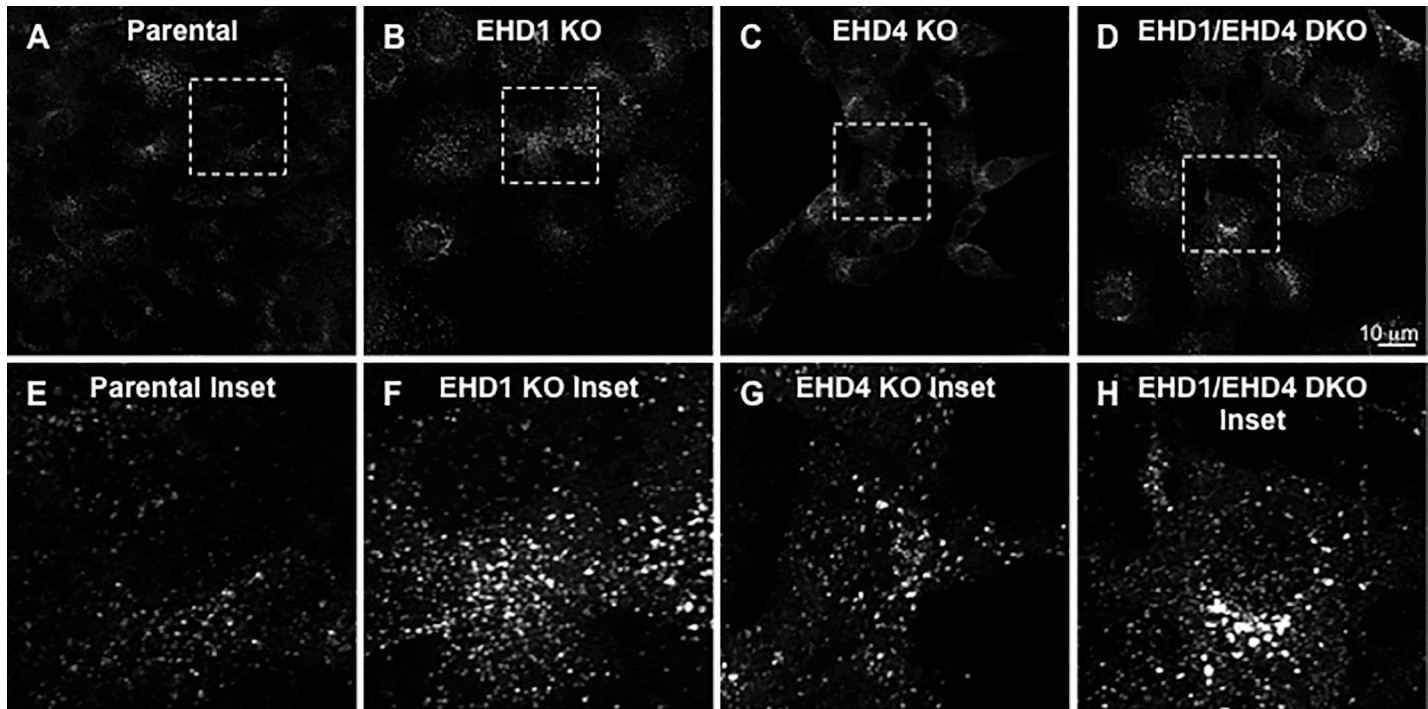

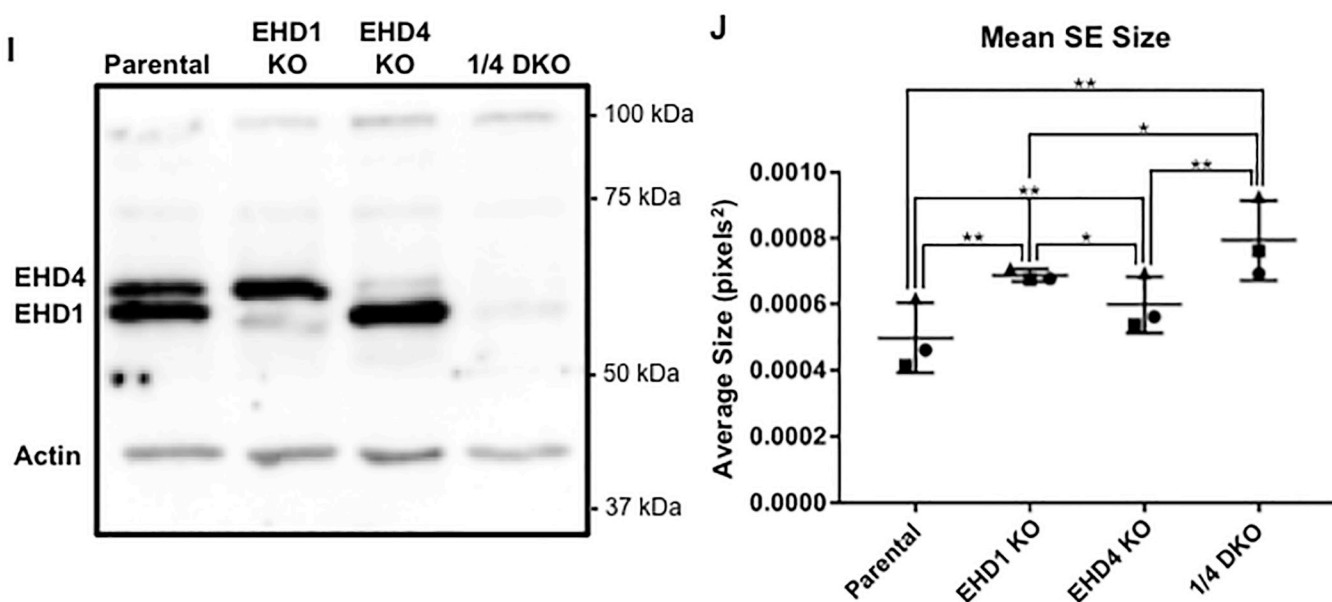

**Fig 3. EHD1 and EHD4 coordinately control endosome size.** *A-H*, Representative micrographs and insets for parental NIH3T3 cells (Parental; *A* and inset in *E*), EHD1 knock-out NIH3T3 cells (EHD1 KO; *B* and inset in *F*), EHD4 knock-out NIH3T3 cells (EHD4 KO; *C* and inset in *G*), and EHD1/EHD4 double knock-out cells (EHD4 DKO; *D* and inset in *H*). Parental NIH3T3 and CRISPR/Cas9 gene-edited NIH3T3 cells lacking either EHD1 (EHD1 KO), EHD4 (EHD4 KO), or both EHD1 and EHD4 (EHD1/EHD4 DKO) were fixed and immunostained with antibodies to EEA1, and then imaged by confocal microscopy. *I*, Immunoblot showing reduced EHD1 expression in EHD1 KO cells, reduced EHD4 expression in EHD4 KO cells and reduced EHD1 and EHD4 expression in EHD1/EHD4 DKO (1/4 DKO) cells. *J*, Graph depicting mean EEA1-labeled endosome size in parental and KO cells. Individual experiments were performed 3 times. Error bars denote standard deviation and p-values were determined by one-way ANOVA for individual experiments using a post-hoc Tukey HSD calculator to determine significance. A consensus p-value was then derived as described in the Materials and methods to assess significant differences between samples from the 3 experiments. Micrographs are representative orthogonal projections from three independent experiments, with 10 sets of z-stacks collected for each treatment per experiment. Bar, 10 μm. Consensus p-values from Tukey HSD: *p = 0.003, **p = 0.001.

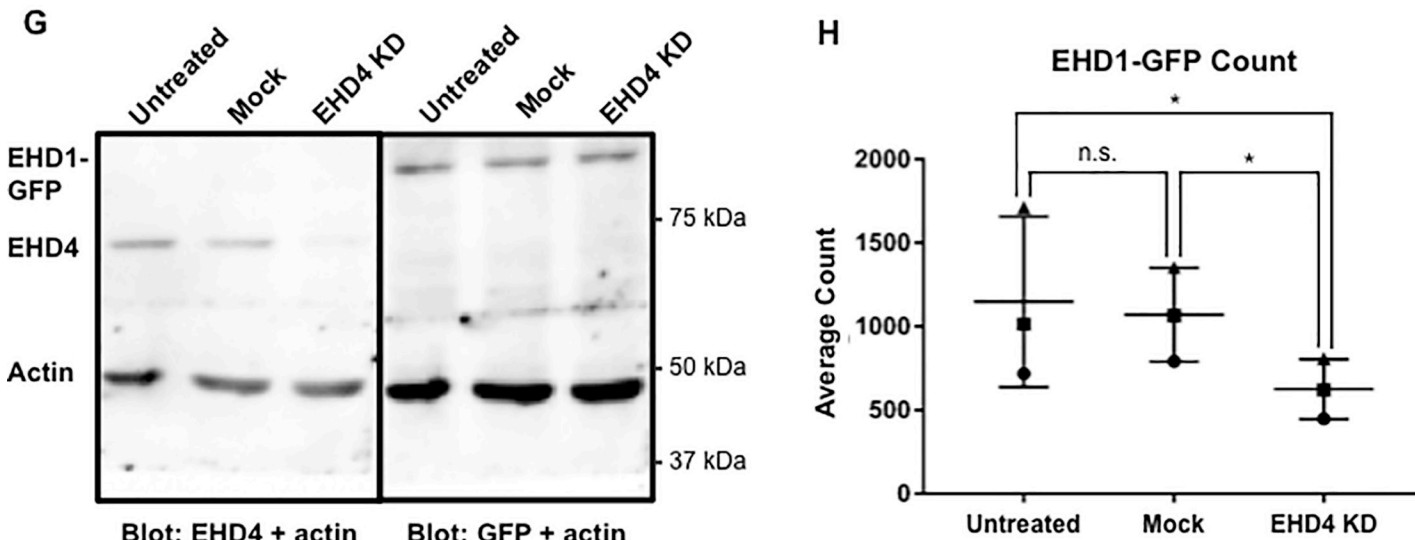

**Fig 4. Reduced EHD1 recruitment to endosomes upon EHD4 knock-down.** *A-F*, Representative micrographs and insets depicting EHD1-GFP recruitment to endosomes in untreated (*A* and inset in *D*), mock-treated (*B* and inset in *E*), and EHD4 knock-down (*C* and inset in *F*) cells. CRISPR/Cas9 gene-edited NIH3T3 cells expressing endogenous levels of EHD1 with GFP fused to the C-terminus (EHD1-GFP) were either untreated, mock-treated with transfection reagent, or transfected with EHD4 siRNA for 72 h. Cells were then incubated with anti-LRP1 antibody (30 min on ice, 30 min at 37˚C), fixed, and imaged via confocal microscopy. *G,*

Immunoblot showing reduced EHD4 (but not EHD1-GFP) expression in EHD1-GFP cells, with actin used as a loading control. The nitrocellulose filter paper was then stripped and immunoblotted with anti-GFP to show EHD1-GFP expression upon EHD4 loss. *H*, Graph depicting the mean count of EHD1-labeled endosomes in untreated, mock-treated and EHD4-depleted cells. Individual experiments were performed 3 times. Error bars denote standard deviation and p-values were determined by one-way ANOVA for individual experiments using a post-hoc Tukey HSD calculator to determine significance. A consensus p-value was then derived as described in the Materials and methods to assess significant differences between samples from the 3 experiments. Micrographs are representative orthogonal projects from three independent experiments, with 10 sets of z-stacks collected for each treatment per experiment Bar, 10 μm. n.s., not significant (consensus p > 0.5). Consensus p-values from Tukey HSD: *p < 0.00001.

Finally, we assessed whether MICAL-L1 is required for EHD1 recruitment and SE fission (Fig 8). As noted, unlike Rabenosyn-5 and Syndapin2, MICAL-L1 bound only to EHD1 and not EHD4 (Fig 5). Nonetheless, upon MICAL-L1 knock-down (validated by immunoblotting in Fig 8M), significantly less EHD1 was recruited to endosomal membranes (compare Fig 8H and 8K and 8B and 8E; quantified in O). Furthermore, EEA1-labeled SE size was enhanced in

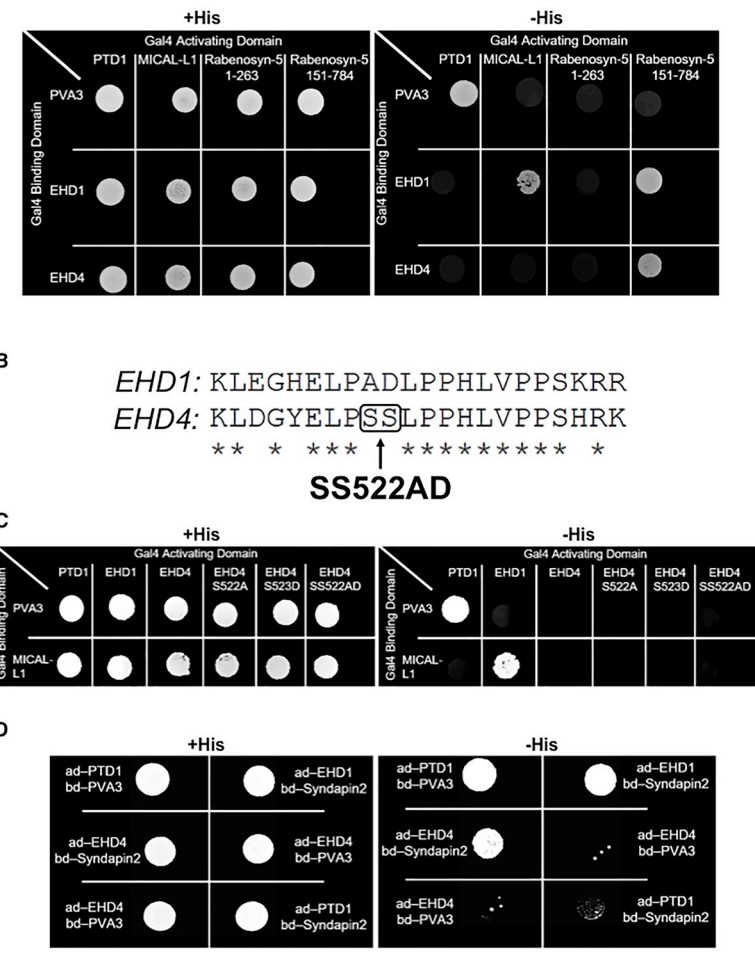

**Fig 5. EHD4 interacts with sorting endosome resident proteins.** *A*, Yeast two-hybrid colony growth demonstrating interactions between both EHD1 and EHD4 with Rabenosyn-5, and between EHD1 and MICAL-L1. Two Rabenosyn-5 constructs were utilized: Rabenosyn-5 151–784 contains 5 Asparagine-Proline-Phenylalanine (NPF) motifs, whereas Rabenosyn-5 1–263 is devoid of NPF motifs. *B*, Schematic illustration depicting residue homology between a region within the EH-domains of EHD1 and EHD4. *C*, Yeast two-hybrid colony growth assessing the interactions between either EHD1, EHD4, or EHD4 mutants with MICAL-L1. *D*, Yeast two-hybrid assay depicting an interaction between either EHD1 or EHD4 with Syndapin2. ad; activation domain, bd; binding domain.

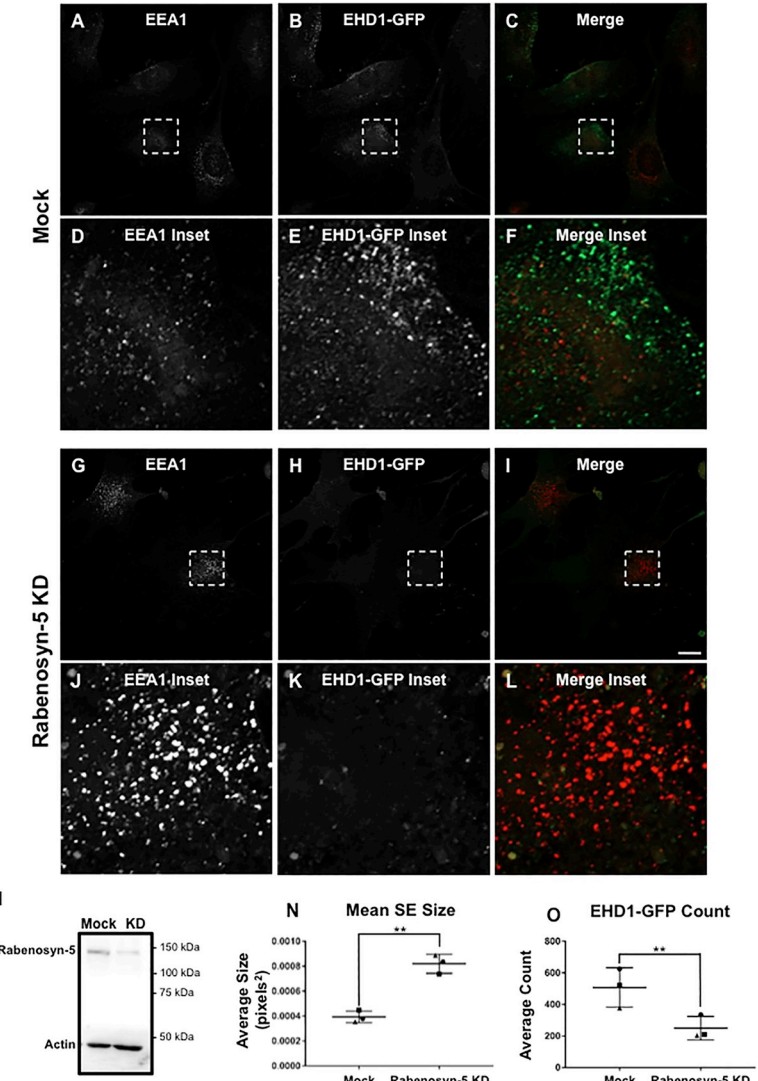

**Fig 6. Increased sorting endosome size and decreased EHD1 recruitment upon Rabenosyn-5 knock-down.** *A-L*, Representative micrographs and insets depicting EEA1-labeled endosomes and EHD1-GFP in mock-treated and Rabenosyn-5 knock-down cells. CRISPR/Cas9 gene-edited NIH3T3 EHD1-GFP cells were either mock-treated with transfection reagent (*A-F*) or treated with Rabenosyn-5 siRNA (*G-L*) for 72 h. Cells were then incubated with anti-LRP1 antibody (30 min on ice, 30 min at 37˚C), fixed and immunostained using anti-EEA1, and imaged by confocal microscopy. *M*, Immunoblot showing reduced Rabenosyn-5 expression in EHD1-GFP NIH3T3 cells. *N*, Graph depicting mean endosome size of mock-treated and Rabenosyn-5 knock-down cells. *O*, Graph depicting EHD1 recruitment to endosomes in mock-treated and Rabenosyn-5 knock-down cells. Error bars denote standard deviation and p-values were determined by independent two-tailed t-test, with significance derived from consensus p-values from the 3 experiments. Micrographs are representative orthogonal projections from three independent experiments, with 10 sets of z-stacks collected for each treatment per experiment. Bar, 10 μm. **p < 0.00001.

the MICAL-L1 knock-down cells compared to mock-treated cells (compare Fig 8G and 8J to 8A and 8D; quantified in 8N). Despite the inability of EHD4 to interact with MICAL-L1, these findings may result from the tight interaction between MICAL-L1 and Syndapin2, since degradation of either protein occurs in the absence of its binding partner [31]. Nonetheless, these data indicate that despite being unable to interact directly with EHD4, MICAL-L1 also serves as a potential docking/recruiting site for EHD dimers at the SE membrane.

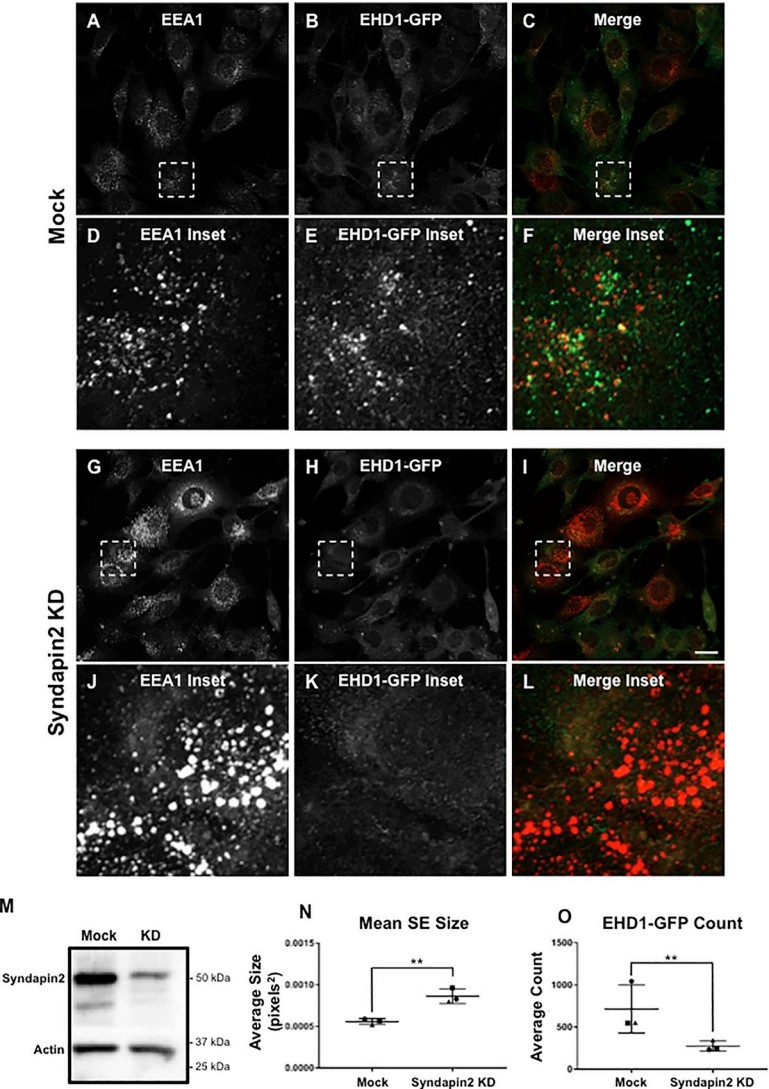

**Fig 7. Increased sorting endosome size and decreased EHD1 recruitment upon Syndapin2 knock-down.** *A-L,*
Representative micrographs and insets depicting EEA1-labeled endosomes and EHD1-GFP in mock-treated and
Syndapin2 knock-down cells. CRISPR/Cas9 gene-edited NIH3T3 EHD1-GFP cells were either mock-treated with
transfection reagent (*A-F*) or treated with Syndapin2 siRNA (*G-L*) for 72 h. Cells were then incubated with anti-LRP1
antibody (30 min on ice, 30 min at 37˚C), fixed and immunostained using anti-EEA1, and imaged by confocal
microscopy. *M,* Immunoblot showing reduced Syndapin2 expression in EHD1-GFP NIH3T3 cells. *N,* Graph depicting
mean endosome size of mock-treated and Syndapin2 knock-down cells. *O,* Graph depicting EHD1 recruitment to
endosomes in mock-treated and Syndapin2 knock-down cells. Error bars denote standard deviation and p-values were
determined by independent two-tailed t-test, with significance derived from consensus p-values from the 3
experiments. Micrographs are representative orthogonal projections from three independent experiments, with 10 sets
of z-stacks collected for each treatment per experiment. Bar, 10 μm. **p < 0.00001.

EHD4 is perhaps the most poorly characterized of the C-terminal EHD family of proteins.
Although a variety of physiologic functions have been proposed for it, including within cardiac
and kidney cells [66–68], testis development [69], neurons [43, 70, 71] and the extracellular
matrix [72], to date its mechanistic function in endocytic membrane trafficking has not been
addressed extensively. In this study, we have characterized EHD4's ability to hetero- and
homo-oligomerize and identified several EHD4 interaction partners that also interact with
EHD1.

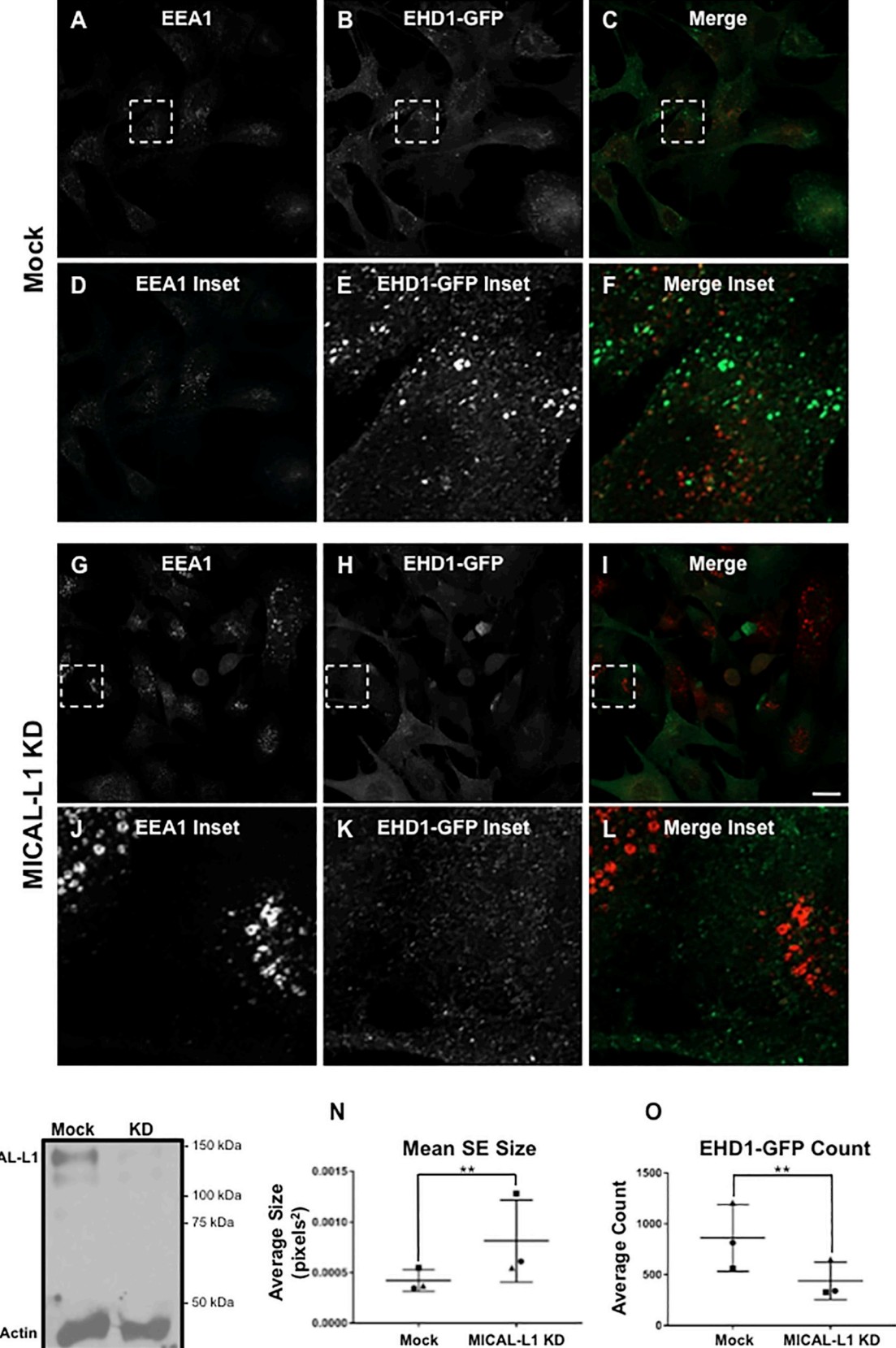

**Fig 8. Increased sorting endosome size and decreased EHD1 recruitment upon MICAL-L1 knock-down.** *A-L*, Representative micrographs and insets depicting EEA1-labeled endosomes and EHD1-GFP in mock-treated and MICAL-L1 knock-down cells. CRISPR/Cas9 gene-edited NIH3T3 EHD1-GFP cells were either mock-treated with transfection reagent (*A-L*) or treated with

MICAL-L1 siRNA (*G-L*) for 72 h. Cells were then incubated with anti-LRP1 antibody (30 min on ice, 30 min at 37˚C), fixed and immunostained using anti-EEA1, and imaged by confocal microscopy. *M*, Immunoblot showing reduced MICAL-L1 expression in EHD1-GFP NIH3T3 cells. *N*, Graph depicting mean endosome size of mock-treated and MICAL-L1 knock-down cells. *O*, Graph depicting EHD1 recruitment to endosomes in mock-treated and MICAL-L1 knock-down cells. Error bars denote standard deviation and p-values were determined by independent two-tailed t-test, with significance derived from consensus p-values from the 3 experiments. Micrographs are representative orthogonal projections from three independent experiments, with 10 sets of z-stacks collected for each treatment per experiment. Bar, 10 μm. $^{**}$p < 0.00001.

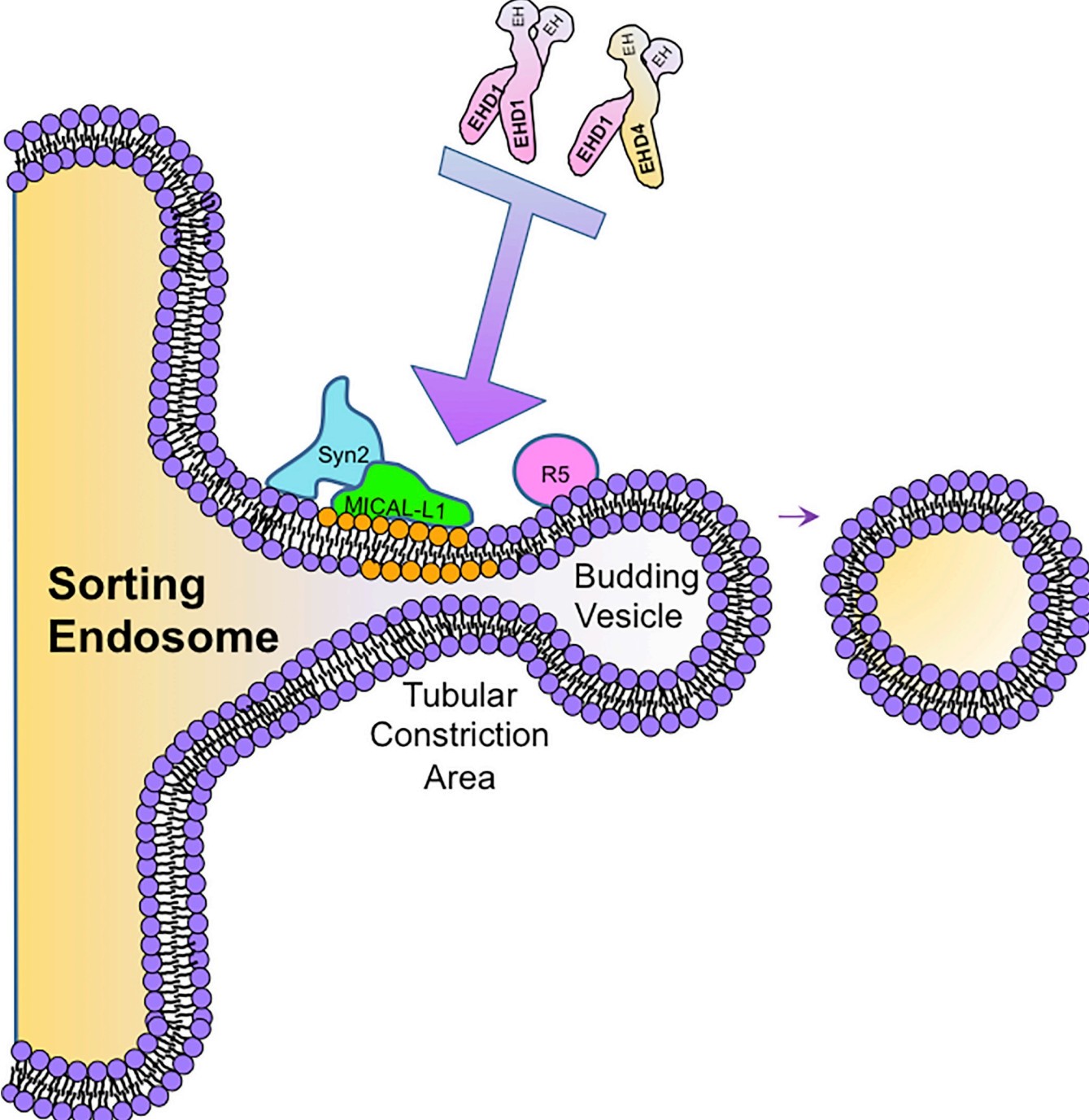

**Fig 9. Model depicting potential mechanisms for EHD1 endosomal recruitment.** Syn2, Syndapin2; R5, Rabenosyn-5; EHD1, Eps15 Homology Domain Protein 1; EHD4, Eps15 Homology Domain Protein 4; EH, Eps15 Homology Domain.

Although the precise stoichiometry of EHD dimers and interaction partners on SE remains unclear, several possibilities exist (Fig 9). For example, EHD1-EHD4 hetero-dimers could either bind to Syndapin2-MICAL-L1 complexes, with EHD1 interacting directly with MICAL-L1 and EHD4 interacting with one of the Syndapin2 NPF motifs. Alternatively, the hetero-dimeric EHD1-EHD4 proteins could dock by binding two adjacent Rabenosyn-5 proteins on the cytoplasmic face of SE membrane, bound to phosphatidylinositol-3-phosphate via its **F**ab 1, **Y**OTB, **V**ac 1, and **E**EA1 (FYVE) domain [73]. On the other hand, EHD1 homo-dimers can bind MICAL-L1-Syndapin2 complexes or Rabenosyn-5 in an unrestricted manner, facilitating recruitment.

Whereas previous studies have addressed the localization of EHD4 to SE [32, 33], these studies predated the concept of EHD1 as a major endosomal fission protein [19, 22, 23, 45, 74]. Moreover, while previous studies demonstrated that EHD4 was in part recruited to SE [32], those studies did not quantify the degree of recruitment, and in our current study we observe significantly impaired recruitment of EHD1 to SE. Our current study supports a role for EHD4 as a dimeric partner with EHD1, facilitating its recruitment to SE and thus similarly implicating EHD4 as a protein intimately connected to the SE fission machinery. These findings are significant, especially since a role for EHD proteins was initially hypothesized in membrane curvature rather than directly in fission [75].

While the stoichiometry of EHD dimers and the precise mode of their recruitment will require further examination, our study helps clarify the complex mechanisms by which EHD1 is recruited to SE to carry out fission and facilitate recycling. We have demonstrated a role for EHD4 in the recruitment of EHD1 to SE, along with at least 3 SE proteins that interact with either EHD1, or both EHD1 and EHD4, namely Rabenosyn-5, Syndapin2 and MICAL-L1. Whether additional SE proteins are also involved in docking EHD dimers on SE remains to be determined.

## Supporting information

**S1 Fig. Partial co-localization between EHD1 and EHD4.** *A-F*, HA-EHD4 was transfected into CRISPR/Cas9 gene-edited cells expressing endogenous levels of EHD1-GFP on coverslips, fixed and stained with primary antibodies against HA and secondary Alexa-568 antibodies and imaged to detect HA-EHD4 (red; *A* and inset in *D*), EHD1-GFP (green; *B* and inset in *E*) and then merged to show both channels (*C* and inset in *F*). *G*, Immunoblot shows expression of the correct-sized HA-EHD4 band at ~65 kDa. The Pearson's Coefficients were calculated with the NIH ImageJ plugin JACoP, and averaged to provide a value of 0.677 (~68%) with a standard deviation of 0.048.
(TIFF)

**S2 Fig. EHD1 immunoprecipitates Rabenosyn-5 in the absence of EHD4.** *A*, NIH3T3 parental cells were grown on a culture dish, pelleted, lysed and either subject directly to SDS PAGE (lysate; right lane), or first immunoprecipitated with beads only (control; left lane) or with anti-EHD1 coupled beads (middle lane) before immunoblotting with anti-Rabenosyn-5 and anti-EHD1. *B and C*, CRISPR/Cas9 gene-edited NIH3T3 cells knocked out for EHD4 were grown on a culture dish, pelleted, lysed and either subject directly to SDS PAGE (lysate; right lane), or first immunoprecipitated with beads only (control; left lane) or with anti-EHD1 coupled beads (middle lane) before immunoblotting with anti-Rabenosyn-5 and anti-EHD1. *C* is a darker exposure of the immunoblot depicted in *B*.
(TIFF)

**S3 Fig. Full-sized gels and repetitions from all of the experiments in the manuscript.**
(PDF)

## Acknowledgments

The content is solely the responsibility of the authors and does not necessarily represent the official views of the National Institutes of Health or any funding agency. The authors wish to thank Eylon Caplan for his discussions and assistance in adopting and modifying a robust consensus p-value test.

## Author Contributions

**Conceptualization:** Naava Naslavsky, Steve Caplan.

**Data curation:** Tyler Jones.

**Formal analysis:** Tyler Jones.

**Funding acquisition:** Steve Caplan.

**Investigation:** Tyler Jones, Naava Naslavsky.

**Methodology:** Tyler Jones.

**Supervision:** Naava Naslavsky, Steve Caplan.

**Validation:** Tyler Jones, Steve Caplan.

**Writing – original draft:** Tyler Jones, Steve Caplan.

**Writing – review & editing:** Tyler Jones, Naava Naslavsky, Steve Caplan.

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
