## [Decision Letter · Decision Letter 0]

7 Apr 2020

PONE-D-20-06631

Eps15 Homology Domain Protein 4 (EHD4) is required for Eps15 Homology Domain Protein 1 (EHD1)-mediated endosomal recruitment and fission

PLOS ONE

Dear Dr. Caplan,

Thank you for submitting your manuscript to PLOS ONE. After careful consideration, we feel that it has merit but does not fully meet PLOS ONE’s publication criteria as it currently stands. Therefore, we invite you to submit a revised version of the manuscript that addresses the points raised during the review process.

We would appreciate receiving your revised manuscript by May 22 2020 11:59PM. To enhance the reproducibility of your results, we recommend that if applicable you deposit your laboratory protocols in protocols.io, where a protocol can be assigned its own identifier (DOI) such that it can be cited independently in the future. For instructions see: http://journals.plos.org/plosone/s/submission-guidelines#loc-laboratory-protocols

We look forward to receiving your revised manuscript.

Kind regards,

Ruben Claudio Aguilar, PhD

Academic Editor

PLOS ONE

"The authors gratefully acknowledge funding support from the National Institutes of General Medical Sciences (NIGMS) (1R01GM123557). The authors thank the University of Nebraska Medical Center Advanced Microscopy Core Facility, which receives partial support from NIGMS: INBRE P20 GM103427 and COBRE P30 GM106397 grants, as well as support from the National Cancer Institute for The Fred & Pamela Buffett Cancer Center Support Grant P30 CA036727, and the Nebraska Research Initiative."

"S.C. was supported by R01 GM123557 from the National Institutes of General Medical Sciences at the National Institutes of Health. The funder had no role in study design, data collection, analysis, decision to publish or manuscript preparation."

Reviewers' comments:

Reviewer's Responses to Questions

**Comments to the Author**

1. Is the manuscript technically sound, and do the data support the conclusions?

Reviewer #1: Partly

Reviewer #2: Partly

2. Has the statistical analysis been performed appropriately and rigorously? 

Reviewer #1: Yes

Reviewer #2: No

3. Have the authors made all data underlying the findings in their manuscript fully available?

Reviewer #1: Yes

Reviewer #2: Yes

4. Is the manuscript presented in an intelligible fashion and written in standard English?

Reviewer #1: Yes

Reviewer #2: Yes

5. Review Comments to the Author

Reviewer #1: The study aimed to unveil the role of EHD4 in the fission process that occurs in sorting endosomes (SE). The interaction between EHD1 and EHD4 was reported previously in Sharma et al., 2008. This work unfortunately has not clearly elucidated further the role of EHD4 in the process. Few key questions need to be answered for acceptance of this manuscript.

Specific comments:

1. Fig. 1A: Please address the extra bands seen in the Input blot.

2. Fig. 1B seems incomplete without showing the input levels for EHD1.

3. Fig. 1C: What is the minimal region required for interaction between EHD1 and EHD4? If only the ATP-binding G-domain is required, why do we not see nay interaction when 1-309 fragment is used? Why not include a mutant which lacks completely the EH domain to test its role in the interaction? Seems from your results that something between the residues in EHD1 between 309-439 is required for binding. Kindly discuss this in the text.

4. Fig. 2G: What is the level of EHD1 in these cells? I think it is critical to address this given the discrepancy in the average size of endosome between EDH4 KD and KO in Fig. 2H and Fig. 3H

5. Fig. 3G: How many times was this blot repeated? Are actin levels always much lower than EHD1 and EHD4 levels.

6. Do EHD1 and EHD4 co-localize in cells? If yes, on what compartment and under what conditions? For example, what happens to their co-localization upon internalization of LRP1?

7. The above question will help clarify results from Figure 4. What compartment is EHD1 recruited to? Are the number of these compartments affected or only the recruitment of EHD1 to these structures?

8. Fig. 6-8: These figures do not reveal the role of EHD4 in recruitment of EHD1 to the SE for fission. What is the overlap, if any, between EEA1 and EHD1-GFP? How is this altered in the KD of Rabenosyn-5, Syndapin-2 and MICAL- L1? What is the localization of EHD4 in mock versus the KD of Rabenosyn-5, Syndapin-2 and MICAL- L1? Is the co-localization between EHD1 and EHD4 affected in KD of Rabenosyn-5, Syndapin-2 and MICAL- L1?

9. If EHD4 plays a role in the recruitment of EHD1 to SE via Rabenosyn-5, Syndapin-2 and MICAL- L1, an immunoprecipitation of EHD1 with these proteins in the presence/ or absence of EDH4 will clarify this. Without these experiments, the role of EHD4 in the recruitment of EHD1 via interaction with partners, since they have common partners, is not convincing.

Reviewer #2: This manuscript by Jones et al explores the mechanisms by which EHD4 (a C-terminal EH domain-containing ATPase) contributes to endosome tubulation and fission. These authors and other groups previously found that EHD4 can vesiculate membranes, that it controls endosome size and sorting function, that it binds to its better-characterized homolog EHD1, and that EHD4 is required for normal EHD1 localization to endosomes (George 2007, Sharma, 2008, Cai 2013).

In this manuscript, the authors further define the functional relationship between EHD1 and EHD4. They narrow down the domains through which EHD4 heterodimerizes with EHD1. They extend on the previous cellular studies by using new CRISPR EHD1 and EHD4 single and double knockout cell lines and an EHD1-GFP knockin cell line, showing that EHD4 is involved in EHD1 recruitment to tubular structures. Finally, they find a role for a variety of EHD1/EHD4 binding partners in this process including Rabenosyn-5, Syndapin-2 and MICAL-L1 (though they cannot at present distinguish if these effects are via EHD4 or EHD1 interactions). The data presented opens up new mechanisms for EHD4 function and the authors hypothesize on possible mechanisms for how EHD4 promotes EHD1 mediated fission of sorting intermediates from the endosome. Overall, this work adds more mechanistic understanding of the role of EDH4 to the puzzle of molecular players involved in endosomal fission. To improve the paper, the authors should provide additional clarification and explanations for their rationale and methods as outlined below, and add some more context to place their results in the literature.

Major Points

1. I have several points relating to putting the manuscript in the context of the field.

a. The authors should note more explicitly which of the figures replicate theirs and others’ previous findings (e.g. co-IP of overexpressed EHD1 and EHD4 and endogenous EHD4 by EHD1, some of the characterization of EHD protein domain interactions (e.g. V203P), endosome enlargement upon EHD4 depletion (George 2007, Sharma 2008)), MICAL-L1 being required for EHD1 recrtuiment (Sharma 2009) to put the work in the context of the literature and most importantly to guide the reader to what is new.

b. In particular, the authors previously reported (Sharma 2008) that EHD4 KD led to redistribution of EHD1 to large early endosomes marked with Rab5, whereas in this report, using LRP1-labeled compartments to mark the sorting endosome, they see a “loss” of EHD1 from structures (Fig 4). Can they please introduce this previous result in the text and discuss the difference in these assays (and see point below regarding overall EHD1 levels in these cells)?

c. The introduction could be expanded to include more clarification on the connections between the known players involved in budding vesicle formation. Further explanation on known binding partners (Rabenosyn-5, MICAL-L1, syndapin2) would increase the salience of identifying EHD4s interaction with these proteins. Also making a connection between known regulators of vesicle fission (WASH, Retromer, FERARI) and EHD1 would enhance the introduction/discussion.

2. I have several points relating to experimental design and statistical analysis

a. The authors should justify their rationale for choosing one-tailed t-tests for significance as opposed to other statistical methods, particularly for data sets including more than two conditions. Even if a change in a particular direction is expected, it would be more rigorous for the authors to use a two-tailed test, and even better to use ANOVA as opposed to a t-test for experiments with three or more conditions.

b. The authors should justify the use of experiment (n=3) rather than cells within a representative experiment as their biological replicates, and comment on the spread of the data, particularly with respect to endosomal size. In Figure 2 and 3 in particular, one of the experimental averages is much lower than the others. Does the lower point for each condition correspond to the same experiment?

3. I have several points relating to experimental interpretation

a. The double KD analysis needs clarification. The effect of each of EHD1 and EHD4 single KO on endosome size was previously published, but EHD1 KO is not included in the current experiments for comparison. It is especially relevant to understand how the increase in sorting endosome size in the EHD1 and EHD4 double knock down compares to the EHD1 single knock down. How does the single EHD1 KD compare to EHD4 KOD and EHD1/EHD4 double KD? Does this analysis indicate that in addition to a shared function through heterodimerization, that EHD1 or EHD4 also have functions independent of each other?

b. In Figure 4 was there an overall decrease in EHD1-GFP signal or just a decrease in the localization to vesicles and tubules? If the overall EHD1-GFP signal is decreased, is possible that the decrease in localization to vesicles and tubules is due to an overall decrease in EHD1-GFP?

Minor Points

1. If possible, IPs should show input (on the same blot without lane cropping, with an indication of fraction loaded relative to IP) to show how much of lysate EHD4 immunoprecipitates with EHD1.

2. The authors should add clarification and rationale for choosing the various truncation mutations of EHD1 used to identify EHD1-EHD4 interactions (Figure 1B) and clarification of the LRP1 uptake assay and how this method is used to detect increases in EHD1 localization to vesicles and tubes (Figure 4). This would increase the accessibility of the paper to a wider audience.

3. The authors should comment in the text on why gene edited cells might exhibit a much more subtle EHD4 endosome enlargement phenotype than the KD cells.

4. Authors should describe secondaries and method used to detect immunoblots so the reader can understand if there may be IgG bands.

5. Authors should list the name of the FIJI macro/plugin used for particle analysis, and explain the rationale/use of different brightness settings.

6. PLOS authors have the option to publish the peer review history of their article (what does this mean?). If published, this will include your full peer review and any attached files.

Reviewer #1: No

Reviewer #2: No

---

## [Author Response · Author response to Decision Letter 0]

20 Aug 2020

Point-by-Point Rebuttal:

Reviewer #1: The study aimed to unveil the role of EHD4 in the fission process that occurs in sorting endosomes (SE). The interaction between EHD1 and EHD4 was reported previously in Sharma et al., 2008. This work unfortunately has not clearly elucidated further the role of EHD4 in the process. Few key questions need to be answered for acceptance of this manuscript.

Specific comments:

1. Fig. 1A: Please address the extra bands seen in the Input blot.

We apologize for not previously marking the 50 kDa band observed in the HeLa lysate lane as “non-specific,” but we have now done so. The only other “extra” band is the weak one observed above the EHD4, which could be a phosphorylated form, but at present this is unknown. 

2. Fig. 1B seems incomplete without showing the input levels for EHD1.

We now provide the input levels for EHD1 for the right-panel of Fig. 1A as requested. 

3. Fig. 1C: What is the minimal region required for interaction between EHD1 and EHD4? If only the ATP-binding G-domain is required, why do we not see nay interaction when 1-309 fragment is used? Why not include a mutant which lacks completely the EH domain to test its role in the interaction? Seems from your results that something between the residues in EHD1 between 309-439 is required for binding. Kindly discuss this in the text.

The minimal region required for interaction between EHD1 and EHD4 is within residues 1-439, meaning that from amino acid 439 to the C-terminus at residue 534 (the EH-domain) the residues are superfluous. Unfortunately, the arrows indicating amino acid positions in Fig. 1B were misaligned and we apologize for the confusion; this has now been corrected. The reviewer is correct that our data suggest that the region between residues 309-439 may be responsible for binding, and we speculate that in addition, the V203P mutant, which weakens EHD1 homo-oligomerization, has an even more deleterious effect on the EHD1-EHD4 heterodimers and their interaction. We have now amended the text to reflect these points. 

4. Fig. 2G: What is the level of EHD1 in these cells? I think it is critical to address this given the discrepancy in the average size of endosome between EDH4 KD and KO in Fig. 2H and Fig. 3H

The reviewer has raised a good point, and we now provide an immunoblot showing EHD1 levels upon EHD4 acute KD, and have commented on the slight decrease in EHD1 expression under these conditions. With regard to the discrepancy in endosome size between acute EHD4 siRNA knock-down and chronic CRISPR/CAS9 gene-edited EHD4 knock-out cells, we respectfully speculate that long-term chronic knock-out is likely met with mechanisms of compensation, thus the effects are less dramatic. We have amended the text to discuss and highlight this possibility.

5. Fig. 3G: How many times was this blot repeated? Are actin levels always much lower than EHD1 and EHD4 levels.

The immunoblots and experiments in Fig. 3 were repeated 3 times (note a new immunoblot has been included with both EHD1 and EHD4 KO in addition to EHD1/EHD4 KO); this information has now been included in the figure legend. With regard to the actin levels observed—they are not “lower” than the EHD levels, but rather result from a lower concentration of primary and secondary antibodies so that actin, the most highly expressed protein in the cell, can be viewed in the same exposure as the EHD proteins. 

6. Do EHD1 and EHD4 co-localize in cells? If yes, on what compartment and under what conditions? For example, what happens to their co-localization upon internalization of LRP1?

This is an excellent question, but unfortunately there are no commercial antibodies available that specifically mark EHD4 by immunofluorescence (without cross-reacting with EHD1). A decade ago, we had prepared rabbit serum in-house to selectively detect EHD4, but once the serum had been used up, we have since been unable to generate an effective new anti-EHD4 serum. To address this important question, we first sought to generate new EHD4 antibodies. While the new antibodies are excellent for immunoblotting, even following affinity purification they are not suitable for immunostaining experiments. Although we recognize it is not ideal and would prefer to immunostain endogenous EHD proteins, to address the concern we have transfected EHD4 into our CRISPR/Cas9 gene-edited EHD1-GFP expressing cells to determine the degree of co-localization, and provide this new information in the form of new Supplemental Fig. 1. 

7. The above question will help clarify results from Figure 4. What compartment is EHD1 recruited to? Are the number of these compartments affected or only the recruitment of EHD1 to these structures?

We have previously shown that EHD1 is recruited to the early endosome/sorting endosome (EE/SE), tubular recycling endosomes (TRE), as well as to the ciliary pocket. Upon EHD4 depletion, the number of these compartments is slightly decreased (data not shown), as well as the recruitment of EHD1 to these structures. The EE/SE are still present, though presumably due to a loss of fission, there are slightly fewer, yet larger, structures. This slight reduction in number would not account for the significant loss of EHD1 seen in Fig 4.

8. Fig. 6-8: These figures do not reveal the role of EHD4 in recruitment of EHD1 to the SE for fission. What is the overlap, if any, between EEA1 and EHD1-GFP? How is this altered in the KD of Rabenosyn-5, Syndapin-2 and MICAL- L1? What is the localization of EHD4 in mock versus the KD of Rabenosyn-5, Syndapin-2 and MICAL- L1? Is the co-localization between EHD1 and EHD4 affected in KD of Rabenosyn-5, Syndapin-2 and MICAL- L1?

We agree with the reviewer that Figures 6-8 don’t necessarily represent direct recruitment of EHD1 by EHD4 to SE; however, since EHD4 appears to exist preferentially in heterodimeric form with EHD1, these experiments reflect the combined recruitment of the dimer pairs. While it would be ideal to address EHD4 localization with the SE markers, EHD1 nonetheless provides an informative surrogate. We have recently demonstrated that EHD1 can be recruited to EEA1-containing SE (Dhawan et al., 2020, J. Biol. Chem.), and in Figs. 6-8, the failure to recruit EHD1 to these structures in the absence of Rabenosyn-5, Syndapin2 and MICAL-L1 likely reflects impaired recruitment of both EHD1-EHD1 dimers and EHD1-EHD4 dimers. 

9. If EHD4 plays a role in the recruitment of EHD1 to SE via Rabenosyn-5, Syndapin-2 and MICAL- L1, an immunoprecipitation of EHD1 with these proteins in the presence/ or absence of EDH4 will clarify this. Without these experiments, the role of EHD4 in the recruitment of EHD1 via interaction with partners, since they have common partners, is not convincing.

We thank the reviewer for this suggestion and we have performed immunoprecipitations of EHD1 with Rabenosyn-5 in the presence and absence of EHD4, since EHD1 and Rabenosyn-5 display the most robust interaction that can be observed by immunoprecipitation. We provide this new information in the form of new Supplemental Fig. 2. Accordingly, we find that there is no significant difference in interaction. 

Reviewer #2: This manuscript by Jones et al explores the mechanisms by which EHD4 (a C-terminal EH domain-containing ATPase) contributes to endosome tubulation and fission. These authors and other groups previously found that EHD4 can vesiculate membranes, that it controls endosome size and sorting function, that it binds to its better-characterized homolog EHD1, and that EHD4 is required for normal EHD1 localization to endosomes (George 2007, Sharma, 2008, Cai 2013).

In this manuscript, the authors further define the functional relationship between EHD1 and EHD4. They narrow down the domains through which EHD4 heterodimerizes with EHD1. They extend on the previous cellular studies by using new CRISPR EHD1 and EHD4 single and double knockout cell lines and an EHD1-GFP knockin cell line, showing that EHD4 is involved in EHD1 recruitment to tubular structures. Finally, they find a role for a variety of EHD1/EHD4 binding partners in this process including Rabenosyn-5, Syndapin-2 and MICAL-L1 (though they cannot at present distinguish if these effects are via EHD4 or EHD1 interactions). The data presented opens up new mechanisms for EHD4 function and the authors hypothesize on possible mechanisms for how EHD4 promotes EHD1 mediated fission of sorting intermediates from the endosome. Overall, this work adds more mechanistic understanding of the role of EDH4 to the puzzle of molecular players involved in endosomal fission. To improve the paper, the authors should provide additional clarification and explanations for their rationale and methods as outlined below, and add some more context to place their results in the literature.

Major Points

1. I have several points relating to putting the manuscript in the context of the field.

a. The authors should note more explicitly which of the figures replicate theirs and others’ previous findings (e.g. co-IP of overexpressed EHD1 and EHD4 and endogenous EHD4 by EHD1, some of the characterization of EHD protein domain interactions (e.g. V203P), endosome enlargement upon EHD4 depletion (George 2007, Sharma 2008)), MICAL-L1 being required for EHD1 recrtuiment (Sharma 2009) to put the work in the context of the literature and most importantly to guide the reader to what is new.

We appreciate the reviewer’s suggestion and have amended the text to make such clarifications. In particular, we have included a new paragraph (second to last in the Results and Discussion) that addresses these issues directly.

b. In particular, the authors previously reported (Sharma 2008) that EHD4 KD led to redistribution of EHD1 to large early endosomes marked with Rab5, whereas in this report, using LRP1-labeled compartments to mark the sorting endosome, they see a “loss” of EHD1 from structures (Fig 4). Can they please introduce this previous result in the text and discuss the difference in these assays (and see point below regarding overall EHD1 levels in these cells)?

We thank the reviewer for this comment; a major difference between the study by Sharma et al. and our current study is that the former was done under “steady-state” conditions, whereas the current study was done upon LRP1 uptake, which we recently described as a trigger to induce endosomal recruitment of EHD1 (Dhawan et al.). Consistent with our previous study, EHD4 knock-down indeed induces larger SE. In addition, also consistent with our previous study, fewer tubular structures were observed containing EHD1. The reviewer is correct that we previously did observe some EHD1 on the enlarged endosomes, but the former study was not quantified, and in the present study while we do observe EHD1-GFP on the enlarged SE, there is quantifiably less. We have included this explanation in the Discussion (amended to the second last paragraph).

c. The introduction could be expanded to include more clarification on the connections between the known players involved in budding vesicle formation. Further explanation on known binding partners (Rabenosyn-5, MICAL-L1, syndapin2) would increase the salience of identifying EHD4s interaction with these proteins. Also making a connection between known regulators of vesicle fission (WASH, Retromer, FERARI) and EHD1 would enhance the introduction/discussion.

We thank the reviewer for this comment; we have modified the Introduction accordingly to include these points and broaden the overall introduction.

2. I have several points relating to experimental design and statistical analysis

a. The authors should justify their rationale for choosing one-tailed t-tests for significance as opposed to other statistical methods, particularly for data sets including more than two conditions. Even if a change in a particular direction is expected, it would be more rigorous for the authors to use a two-tailed test, and even better to use ANOVA as opposed to a t-test for experiments with three or more conditions.

We have taken the comments regarding our statistical analyses with utmost seriousness, and have made the following changes: 1) We have applied one-way ANOVA where necessary (Figs. 2, 3 and 4) along with the post-hoc Tukey HSD test for significance. 2) We have moved to independent two-tailed t-tests where ANOVA analysis is not needed. 3) Most importantly, given the concerns about assessing statistical significance with biological samples that often present trends, but have wide variations in absolute values between individual experiments (but not between samples within individual experiments), we adopted the use of a Consensus p-value. This method allows researchers to use statistical calculations to derive an “overall” or consensus p-value from 3 or more experiments. The method we adopted is derived from the papers by Folks (reference 41) and Rice (reference 42), and we believe that inclusion of these statistical methods greatly improves our manuscript.

b. The authors should justify the use of experiment (n=3) rather than cells within a representative experiment as their biological replicates, and comment on the spread of the data, particularly with respect to endosomal size. In Figure 2 and 3 in particular, one of the experimental averages is much lower than the others. Does the lower point for each condition correspond to the same experiment?

As discussed in our response to the reviewer’s point above (2a), we have now used a consensus p-value that addresses the very valid concerns of the reviewer. To address biological variations between individual tests, we have designed a modified version of the method described by Folks for deriving a “consensus p-value” to determine the likelihood that the collection of different test/experiments collectively suggests (or refutes) a common null hypothesis, modified from the Liptak-Stouffer method. Such consensus p-values put less emphasis on absolute values, and more on the trends between individual experiments. 

3. I have several points relating to experimental interpretation

a. The double KD analysis needs clarification. The effect of each of EHD1 and EHD4 single KO on endosome size was previously published, but EHD1 KO is not included in the current experiments for comparison. It is especially relevant to understand how the increase in sorting endosome size in the EHD1 and EHD4 double knock down compares to the EHD1 single knock down. How does the single EHD1 KD compare to EHD4 KOD and EHD1/EHD4 double KD? Does this analysis indicate that in addition to a shared function through heterodimerization, that EHD1 or EHD4 also have functions independent of each other?

We appreciate the suggestion by the reviewer and have added the EHD1 KO condition for comparison to EHD4 KO and the EHD1/EHD4 DKO (see revised Fig. 3). 

b. In Figure 4 was there an overall decrease in EHD1-GFP signal or just a decrease in the localization to vesicles and tubules? If the overall EHD1-GFP signal is decreased, is possible that the decrease in localization to vesicles and tubules is due to an overall decrease in EHD1-GFP?

The number of EHD1-GFP punctae was decreased upon EHD4 KD. EHD4 siRNA was assessed to ensure that EHD1 protein levels are not impacted. We have included an immunoblot illustrating that EHD1-GFP protein levels are not impacted upon EHD4 KD.

Minor Points

1. If possible, IPs should show input (on the same blot without lane cropping, with an indication of fraction loaded relative to IP) to show how much of lysate EHD4 immunoprecipitates with EHD1.

We now provide the input levels for EHD1 for the right-panel of Fig. 1A as requested. 

2. The authors should add clarification and rationale for choosing the various truncation mutations of EHD1 used to identify EHD1-EHD4 interactions (Figure 1B) and clarification of the LRP1 uptake assay and how this method is used to detect increases in EHD1 localization to vesicles and tubes (Figure 4). This would increase the accessibility of the paper to a wider audience.

We thank the reviewer for helping us to clarify, and have included this in the text.

3. The authors should comment in the text on why gene edited cells might exhibit a much more subtle EHD4 endosome enlargement phenotype than the KD cells.

We have now addressed this issue in the Results and Discussion.

4. Authors should describe secondaries and method used to detect immunoblots so the reader can understand if there may be IgG bands.

We have added additional explanation describing secondary antibodies and immunoblotting. 

5. Authors should list the name of the FIJI macro/plugin used for particle analysis, and explain the rationale/use of different brightness settings.

No macros or plugins were used for particle analysis, only using tools available in the base version. We have added further explanation of the particle analysis and the rationale behind using particular brightness settings. For the co-localization analysis in the new Supplemental Fig. 2, please see the figure legend for the macro/plugins used.

---

## [Decision Letter · Decision Letter 1]

11 Sep 2020

Eps15 Homology Domain Protein 4 (EHD4) is required for Eps15 Homology Domain Protein 1 (EHD1)-mediated endosomal recruitment and fission

PONE-D-20-06631R1

Dear Dr. Caplan,

We’re pleased to inform you that your manuscript has been judged scientifically suitable for publication and will be formally accepted for publication once it meets all outstanding technical requirements.

Kind regards,

Ruben Claudio Aguilar, PhD

Academic Editor

PLOS ONE

Additional Editor Comments (optional):

Reviewers' comments:

Reviewer's Responses to Questions

**Comments to the Author**

1. If the authors have adequately addressed your comments raised in a previous round of review and you feel that this manuscript is now acceptable for publication, you may indicate that here to bypass the “Comments to the Author” section, enter your conflict of interest statement in the “Confidential to Editor” section, and submit your "Accept" recommendation.

Reviewer #1: All comments have been addressed

Reviewer #2: (No Response)

2. Is the manuscript technically sound, and do the data support the conclusions?

Reviewer #1: Yes

Reviewer #2: Yes

3. Has the statistical analysis been performed appropriately and rigorously? 

Reviewer #1: Yes

Reviewer #2: Yes

4. Have the authors made all data underlying the findings in their manuscript fully available?

Reviewer #1: Yes

Reviewer #2: No

5. Is the manuscript presented in an intelligible fashion and written in standard English?

Reviewer #1: Yes

Reviewer #2: Yes

6. Review Comments to the Author

Reviewer #1: (No Response)

Reviewer #2: Overall, we feel our concerns were addressed and we recommend this paper for publication.

Major points

1. The authors did a good job contextualizing their findings in the field as well as with their previous publication.

2. The authors addressed concerns about the statistical tests and recalculated statistical measures for all experiments in the manuscript. However, they do not detail how they “modified” the Folks and Liptak-Stouffer methods. This should be clarified in the methods.

3. The authors added the additional data which we requested. We note that identifying the specific compartment on which EHD1 and EHD4 interact may be complex and is likely beyond the scope of this paper.

The authors addressed all of our minor concerns.

7. PLOS authors have the option to publish the peer review history of their article (what does this mean?). If published, this will include your full peer review and any attached files.

Reviewer #1: No

Reviewer #2: No

---

## [Editor Report · Acceptance letter]

14 Sep 2020

PONE-D-20-06631R1 

Eps15 Homology Domain Protein 4 (EHD4) is required for Eps15 Homology Domain Protein 1 (EHD1)-mediated endosomal recruitment and fission 

Dear Dr. Caplan:

I'm pleased to inform you that your manuscript has been deemed suitable for publication in PLOS ONE. Congratulations! Your manuscript is now with our production department. 

Kind regards, 

on behalf of

Dr. Ruben Claudio Aguilar 

Academic Editor

PLOS ONE